# Unsupervised Learning of Shape Programs with Repeatable Implicit Parts

**Boyang Deng**[1,*]     **Sumith Kulal**[1,*]     **Zhengyang Dong**[1]     **Congyue Deng**[1]

**Yonglong Tian**[2]     **Jiajun Wu**[1]

[1]Stanford University,    [2]MIT,    *equal contributions

## Abstract

Shape programs encode shape structures by representing object parts as subroutines and constructing the overall shape by composing these subroutines. This usually involves the reuse of subroutines for repeatable parts, enabling the modeling of correlations among shape elements such as geometric similarity. However, existing learning-based shape programs suffer from limited representation capacity, because they use coarse geometry representations such as geometric primitives and low-resolution voxel grids. Further, their training requires manually annotated ground-truth programs, which are expensive to attain. We address these limitations by proposing Shape Programs with Repeatable Implicit Parts (ProGRIP). Using implicit functions to represent parts, ProGRIP greatly boosts the representation capacity of shape programs while preserving the higher-level structure of repetitions and symmetry. Meanwhile, we free ProGRIP from any inaccessible supervised training via devising a matching-based unsupervised training objective. Our empirical studies show that ProGRIP outperforms existing structured representations in both shape reconstruction fidelity and segmentation accuracy of semantic parts.

## 1 Introduction

Representing *geometry* and *structure* simultaneously has long been one of the central pursuits of 3D shape modeling algorithms. The capability of preserving geometric details indicates the sufficient shape capacity of an algorithm. Meanwhile, the prediction of plausible structures provides answers to quests for more intelligent shape understanding — What are the parts constituting an object? How are they assembled? What are the relationships among them?

Primitive-based representations have been extensively studied to address such queries. Specifically, numerous shape primitives are designed to construct objects in a bottom-up manner, including the use of oriented boxes [57], ellipsoids [20], convex polytopes [12, 9], superquadrics [43], and invertible implicit functions [45]. Agnostic to the exact primitive used, these models all predict primitives independently, where each primitive accounts for a unique part of the object. Yet, more common in real world are recurrences of similar or even identical geometric parts, e.g., legs of a chair. Consequently, the absence of recurring parts modeling handicaps the advances of these methods towards human-level understanding of object shapes.

To enable the understanding of shape regularities including the repetition and symmetry of parts, a few recent papers have proposed to cast shape reconstruction as prediction and execution of neural shape programs [55, 25]. The designed domain specific language (DSL) of shape programs supports the prediction of repeatable parts. These methods however suffer from two major limitations. First, they often fail to achieve accurate geometry reconstruction due to the use of coarse geometric

36th Conference on Neural Information Processing Systems (NeurIPS 2022).

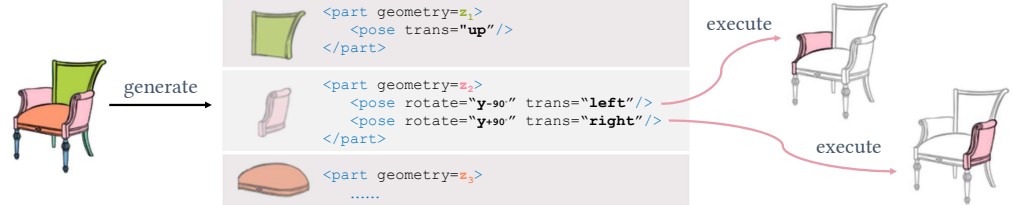

Figure 1: Our method represents an object as a shape program with repeatable implicit parts (ProGRIP). The program has two levels: the top level defines a set of repeatable parts (as latent vector $z_i$) and the bottom level defines all occurrences of each part with varying poses. The joint predictions, i.e., posed parts, are executed as posed implicit functions. Both the generation and the execution of ProGRIP are invariant to the order of predictions at both levels. ProGRIP can be learned without any annotations using our proposed matching-based unsupervised training objective.

representations, such as primitives or low-resolution voxel grids. Second, they often rely on hand-crafted synthetic datasets with ground-truth programs for training or pre-training [55], which prevents them from scaling up to various, unseen object categories — one has to manually annotate shape programs for each new category.

In this work, we address these two fundamental limitations by proposing shape programs with repeatable implicit parts (ProGRIP, Fig. 1), and a method that learns ProGRIP without supervision, illustrated in Fig. 4. As in earlier studies on shape programs, ProGRIP comprises of repeatable parts, rendering the modeling of recurring structures straightforward; unlike them, we now significantly enhance the quality of reconstructed shapes by introducing implicit functions as part representations. Compared with low resolution voxel-based representations, neural implicit representations have recently demonstrated strikingly high fidelity in shape and part modeling [37, 21]. Reformulating shape parts in our programs as neural implicit functions leads to detail-preserving reconstruction.

While earlier studies on shape programs require annotated programs for training, we propose an unsupervised learning objective, matching the oriented bounding boxes of predicted repeatable parts to non-repeatable box-based shape decomposition [64], which allows learning from unannotated shapes. Unlike the reconstruction loss, which confines each part to fitting local shape in the vicinity of its initialized location, our objective encourages repeatable parts to match to object shape elements with similar geometry, even if far from their locations. This leads to unsupervised acquisition of meaningful part decomposition, enabling the learning of ProGRIP for any object category.

We conduct extensive experiments on ShapeNet [6] to validate the effectiveness of ProGRIP on shape reconstruction and semantic part segmentation. Our results demonstrate that ProGRIP consistently preserves higher geometric fidelity than state-of-the-art structured representations on all categories. Meanwhile, it achieves competitive semantic part segmentation accuracy.

In summary, our key contributions are three-fold:

- We propose ProGRIP, introducing implicit functions to represent repeatable parts in shape programs, greatly enhancing their representation capacity of shape programs.
- We develop an unsupervised training objective for learning ProGRIP, relaxing prior requirements of manually designed ground-truth programs.
- We demonstrate the power of ProGRIP representation and learning objective on shape reconstruction, part segmentation, and shape compactness, outperforming state-of-the-art structured representations on multiple categories.

## 2   Related Work

Our method aims to construct primitive-based shape representations with recurring geometric structures. It builds upon neural shape programs and implements primitives as neural implicit functions. Thus, we review the relevant literature in the following three fields:

**Primitive-based shape modeling**.  Most artificial objects naturally exhibit part hierarchy and structural regularities [40]. Primitive-based shape modeling or abstraction is a decent way of modeling

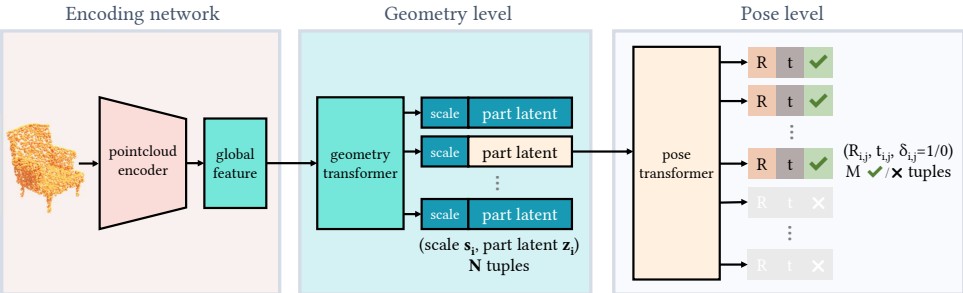

Figure 2: **ProGRIP Generation.** Given a pointcloud, an auto-encoding architecture generates a ProGRIP composed of 2 levels of predictions. At the geometry level, our model predicts a set of $(s_i, z_i)$-pairs as the scales and deep latents of repeatable parts **(middle)**; at the pose level, our model predicts a set of $(t_{i,j}, R_{i,j}, \delta_{i,j})$-triplets as translations, rotations, and existence probabilities **(right)**. Transformers are used at both levels for permutation invariant predictions.

such relations. Some recent works have extended the classic idea of generalized cylinders [48, 3] or geons [2] by describing shapes as a collection of geometric primitives [57, 66, 49, 55, 25, 44, 32, 31, 28, 27, 52, 50, 19, 60, 63, 65]. In particular, Tulsiani et al. [57] and Zou et al. [66] abstracted shapes by minimizing the difference between raw shapes and reconstructions using pre-defined primitives, in an unsupervised fashion. In Sharma et al. [49] and Du et al. [15], 3D CAD programs are inferred from the perceptual input of shapes. In addition to part decomposition, inter-part relations are also widely discussed, Tian et al. [55] defined programs in terms of semantically meaningful parts; Mo et al. [39] built relational graphs upon parts to represent shape structures; Xu et al. [62] and Ritchie et al. [47] studied constructive shape modeling processes. However, these methods usually require detailed structural annotations on real or synthetic data, preventing them from being applied to various unlabeled shape categories.

**Visual program induction**. Programs have been commonly used to capture regularities (e.g. representing repetitions by loops) and have been combined with deep models to describe various visual data, such as 2D graphics [16, 49, 35], 3D shapes [55, 49, 25, 27], scenes [34, 36, 33, 18], and videos [30]. Most of these works [16, 55, 18, 17, 27, 59] follow the paradigm of analysis by synthesis: a recognition network takes as input visual data and outputs programs, which are rendered back to visual data to compare with the original input; such reconstruction error is back-propagated to optimize the recognition network. The rendering can be implemented as either a graphics engine [16] or learned neural models [55]. Our model shares similar spirits with these previous arts, but learns a neural implicit occupancy decoder to render programs into 3D shapes. Additionally, ShapeMOD [26] assumes shape programs as input and proposes to discover macro 'options' to compress them; in contrast, our model aims to infer shape programs and learn to encode parts simultaneously using implicit functions. Also relevant is DeepCAD [61] which utilizes predefined CAD commands to compose 3D shape programs while our model is not restricted to any predefined shape elements and learns the programs from scratch.

**Neural implicit representations**. Neural implicit representations define surfaces of geometry as zero level-sets of volumetric fields parameterized by neural networks [42, 37, 7, 23]. Acquirable from raw shape data [1, 22, 11], neural implicits show excellent performances in capturing high-frequency details at any level [38, 51, 53], even for large-scale scenes [51, 24, 5]. With latent space encoding conditioned on local features [10, 24, 46, 5], poses [14], or parts [13, 41], neural implicits can naturally incorporate geometric and structural priors into its shape representation. A few recent works also reveal category-level structural similarities through implicit reconstruction, such as Genova et al. [20, 21] (via patch-correspondences) and Chen et al. [8] (via part (co-)segmentation), or cross-category geometric similarities [56], with zero or minimal supervision. Recently, Neural Parts [45] also models 3D shapes based on implicit shape elements. Different from our model, Neural Parts treats each element independently hence ignores essential geometric self-similarties in the modeling.

We summarize the key differences between ProGRIP and most relevant prior methods below and empirically compare ProGRIP against them in Sec. 5.

- CubeSeg [64] is a part-based representation that models shapes as a collection of oriented boxes. Each box in the collection is regarded as an independent element with its own shape

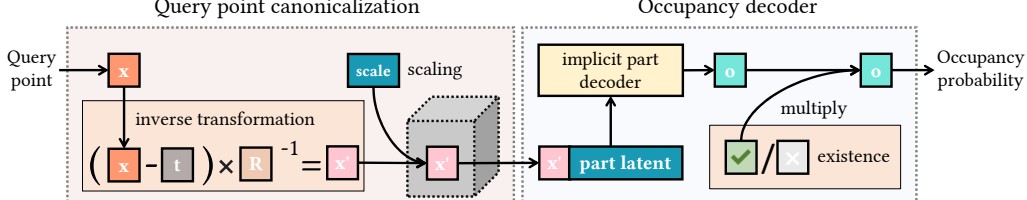

Figure 3: **Posed Implicit Functions.** We execute each posed part (i.e., a $(s_i, z_i, t_{i,j}, R_{i,j}, \delta_{i,j})$ tuple) as a posed implicit function. A posed implicit function constructs an occupancy function $o_{i,j}$ to answer point queries $x$. For each query point $x$, we first canonicalise it using $(s_i, t_{i,j}, R_{i,j})$, then predict its occupancy conditioned on part latent $z_i$, and finally mask it by binarised existance $\hat{\delta}_{i,j}$.

and pose. Contrarily, ProGRIP predicts shape and pose separately and allows the same shape to have multiple copies in various poses. This enables the understanding of self similarity within a shape.

- Shape2Prog [55] predicts programs with repeatable shapes as voxel grids. ProGRIP differs from Shape2Prog in using a higher fidelity neural implicit shape representation and learning programs without supervised pretraining.

- BSP-Net [9] is an implicit representation targeting structured shape understanding. Similar to CubeSeg, it outputs independent shape elements, while ProGRIP recognizes shape similarity among elements.

## 3 ProGRIP

We model shapes using shape programs with repeatable implicit parts, or ProGRIP (Fig. 1). Given an object, we first extract its structural regularity especially repeating geometry via a shape program generator. The generator outputs shape programs with two levels of permutation invariant predictions, parts at the *geometry* level and poses at the *pose* level (Sec. 3.1 and Fig. 2) . The compound set of these predictions are posed parts, which we use to construct object shapes. Each posed part is essentially a posed implicit function (Sec. 3.2 and Fig. 3). Composing all posed parts gives us the overall object shape, which is essentially a spatial occupancy function $\mathcal{O} : \mathbb{R}^3 \rightarrow [0, 1]$ with its surface boundary defined as $\partial\mathcal{O} = \{x \in \mathbb{R}^3 \,|\, \mathcal{O}(x) = 0.5\}$.

### 3.1 Generating ProGRIP

Unlike the sequential programs in Tian et al. [55], we introduce *parallelism* into our program, allowing repeatable parts to be executed simultaneously without a specified order. Specifically, our program generator takes as input a point cloud $\mathbf{X} \in \mathbb{R}^{P \times 3}$ with $P$ points and interprets it as a ProGRIP using an auto-encoding architecture (Fig. 2). A ProGRIP consists of a two-level hierarchy. The top level, namely geometry level, defines a set of repeatable parts as shape building blocks, e.g. the shape of a leg on a chair. The bottom level, namely pose level, predicts a set of poses for each repeatable part from the geometry level, e.g., four copies of the leg shape in four different poses on a chair. Within one pose set are the locations and orientations of all occurrences of the corresponding part in the object (Fig. 1). All predictions at the pose level, joint by the prediction from its parent part, specify all element posed parts (e.g., four legs of a table) that constitute the object shape. Formally, by a slight abuse of notations, we define the hierarchy:

1. **Geometry Level**: $\{(s_i, z_i) \,|\, i = 1, \ldots, N\}$, where $s_i \in \mathbb{R}^3$ describes the axis-aligned scales of the $i$-th repeatable part, $z_i \in \mathbb{R}^d$ encodes the $d$-dimensional deep latent of this part, and $N$ is the number of parts at this level.

2. **Pose Level**: $\{(t_{i,j}, R_{i,j}, \delta_{i,j}) \,|\, j = 1, \ldots, M\}$ for the $i$-th part, where $t_{i,j}$, $R_{i,j}$, and $\delta_{i,j}$ are the translation, rotation, and existence probability of the $j$-th posed occurrence of this part. Note that during program generation, we predict continuous probability as $\delta_{i,j}$ to allow back-propagation of gradients, while during program execution, we binarise $\delta_{i,j}$ into

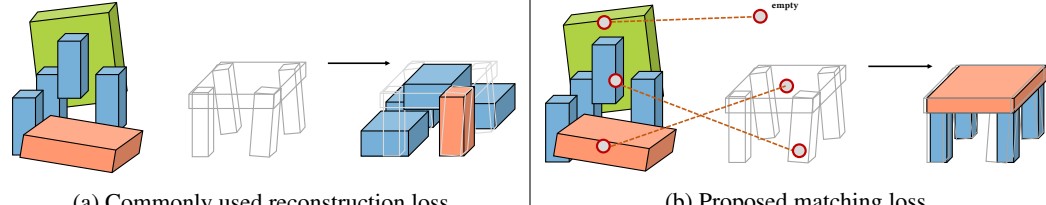


(a) Commonly used reconstruction loss.     (b) Proposed matching loss


Figure 4: **A Demo of Unsupervised Matching Loss.** On a task of fitting repeatable parts to a target shape, starting from the same initialization, the reconstruction loss **(left)** confines each posed parts to their initial local neighborhood and consequently prevents better parts arrangements. Conversely, our matching loss **(right)** match posed parts to local geometry of targets by shapes, thus rescue the part arrangement from the suboptimal local minima in reconstruction loss.

$\hat{\delta}_{i,j} = \mathbb{1}(\delta_{i,j} \geq 0.5)$ and remove occurrences with $\hat{\delta}_{i,j} = 0$ to allow varying numbers of occurrences of the $i$-th part.

In adaptation to the program parallelism, the predictions at both the geometry level and the pose level are **orderless sets**. We employ two nested transformers [58] in our architecture to facilitate permutation invariance at both prediction levels. In particular, the geometry transformer takes in the encoded global feature for the object and predicts the the parts including the scale $s_i$ and a deep part latent $z_i$. Then, we feed each $z_i$ to a pose transformer to predict the $(R_{i,j}, t_{i,j}, \delta_{i,j})$ triplets for each posed occurrence. Combining the pose predictions and their parent part prediction produces a set of $\{(s_i, z_i, t_{i,j}, R_{i,j}, \delta_{i,j})\}$ tuples, each defining a posed part.

### 3.2 Executing ProGRIP

We execute ProGRIP by executing the predicted posed parts as posed implicit functions [37, 42, 7]. Instead of simple polygonal primitives with only a few shape parameters, we propose to use an implicit function $\mathcal{P}(\cdot \,|\, z) : \mathbb{R}^3 \to [0, 1]$ to represent a part. It takes in any point query $x$ and predicts the likelihood of $x$ being inside of a part. $\mathcal{P}$ is the occupancy decoder parameterized by a deep neural network and shared by all parts. $z$ is the part latent from the geometry level. We condition $\mathcal{P}$ on $z$ to obtain a specific implicit function for a part. For a posed part parameterised by the 5-tuple $(s_i, z_i, t_{i,j}, R_{i,j}, \delta_{i,j})$, we construct its posed implicit function $o_{i,j}$ as

$$o_{i,j}(x) = \hat{\delta}_{i,j} \mathcal{P}\left( \left(R_{i,j}^{-1}\left(x - t_{i,j}\right)\right) / s_i \,\middle|\, z_i \right). \tag{1}$$

Note that we binarize $\delta_{i,j}$ to $\hat{\delta}_{i,j}$. As in Fig. 3, given a query point $x$, we first canonicalize its coordinates to $x' = (R_{i,j}^{-1}(x - t_{i,j}))/s_i$. Then we feed $x'$ to $\mathcal{P}$ together with the shape code $z_i$. The binarized existence probability $\hat{\delta}_{i,j}$ is finally multiplied to the predicted occupancy, masking out the non-existing primitives. The eventual output of the program is an object shape as the union of all posed parts. The object occupancy function can therefore be written as

$$\mathcal{O}(x) = \max_{i \in [N]} \max_{j \in [M]} o_{i,j}(x). \tag{2}$$

## 4 Unsupervised Learning of ProGRIP

As our shape modeling consists of two steps, i.e., the generation and execution of ProGRIP, we first learn the ProGRIP generator and then optimize our posed implicit functions. Both optimizations are done without any supervision.

### 4.1 Learning to Generate ProGRIP

Prior works on generating shape programs often rely on hand-crafted synthetic programs to bootstrap the learning [55]. As manually creating programs for numerous object categories is prohibitive, this significantly restricts large-scale applications of these works. To circumvent this limitation, one would

resort to unsupervised training objectives. A trivial design of such objectives is reconstruction losses. Particularly, we only supervise the composition of all elements by the overall reconstruction quality. Despite its decent effectiveness in prior works [12, 9, 57, 64], we show in one demonstrative case (Fig. 4) that it's not applicable when we have repeatable parts. Reconstruction loss encourages each posed part to evolve to the local geometric structure closest to its initialized location. When parts are repeatable, different copies of the same part may appear in irrelevant locations, e.g., one near a chair leg while another close to the chair seat. The local minimal of reconstruction loss consequently traps these copies to its initially assigned neighborhood and suppresses the transformation that would lead to more meaningful assignment of these copies. To address this issue, we devise a matching-based unsupervised objective that supervises a posed part using a local geometry of the object with a similar shape, even if it's far from the current location of this posed part. Practically, we match our predicted posed *repeatable* parts to a *non-repeatable* primitive decomposition of the object, where each primitive serves as a local geometry descriptor.

We acquire the non-repeatable primitive decomposition using the unsupervised box-based algorithm in Yang and Chen [64]. Namely, given pointcloud $\mathbf{X}$, it predicts a set of independent oriented bounding boxes $\{(\overline{\boldsymbol{s}}_k, \overline{\boldsymbol{t}}_k, \overline{\boldsymbol{R}}_k, \overline{\delta}_k) \mid k = 1, \ldots, NM\}$. Here $\overline{\delta}_k$ is a binary value indicating if the $k$-th box exists (1 for present and 0 for absent). If the number of valid boxes is less than $NM$ (i.e., the number of posed parts predicted by our ProGRIP), we pad the remaining entries with all zero. As the subset 4-tuple $\{(\boldsymbol{s}_k, \boldsymbol{t}_k, \boldsymbol{R}_k, \delta_k)\}$ from the 5-tuple specifying a posed part also defines its oriented bounding box, we optimize our predicted primitives by fitting these boxes to $\{(\overline{\boldsymbol{s}}_k, \overline{\boldsymbol{t}}_k, \overline{\boldsymbol{R}}_k, \overline{\delta}_k)\}$. Specifically, we use a box scale loss $\mathcal{L}_s$, as well as two auxiliary losses for the box vertices ($\mathcal{L}_v$) and existence ($\mathcal{L}_e$) to supervise the fitting of boxes. The overall loss is a weighted combination $\mathcal{L}_m$ of these three losses. We name it matching loss. Inspired by Carion et al. [4], we base the computation of all loss terms on a permutation $\rho$ of $NM$ elements, where $\rho(k)$ is the $k$-th element in the permutation. Without loss of generality, we define $\mathcal{L}_s$, $\mathcal{L}_v$, and $\mathcal{L}_e$ for the $k$-th predicted posed part as the following:

**Scale loss $\mathcal{L}_s$.** We first supervise the shape of the posed part using the target local geometry as we seek to find the repeated shape elements in an object. In the context of oriented boxes, we fit the axis-aligned scales of our predicted boxes to their assigned target boxes from the non-repeatable decomposition. We implement it as the complementary of Intersection-over-Union between two axis-aligned origin-centered boxes defined by scales $\boldsymbol{s}_k$ and $\overline{\boldsymbol{s}}_{\rho(k)}$:

$$\mathcal{L}_s(k) = \mathbb{1}_{\{\overline{\delta}_{\rho(k)}=1\}} \left( 1 - \mathrm{IoU}(\boldsymbol{s}_k, \overline{\boldsymbol{s}}_{\rho(k)}) \right). \tag{3}$$

**Vertices loss $\mathcal{L}_v$.** We also require the matched predicted box and the target box to have the same location and orientation by minimizing the $l_1$-distance between the box vertices after posing. Formally, we define the box vertices generated using $(\boldsymbol{R}_k, \boldsymbol{t}_k, \boldsymbol{s}_k)$ as $\boldsymbol{B}_k$ and similarly $\overline{\boldsymbol{B}}_{\rho(k)}$ for $(\overline{\boldsymbol{R}}_k, \overline{\boldsymbol{t}}_k, \overline{\boldsymbol{s}}_k)$. Consequently,

$$\mathcal{L}_v(k) = \mathbb{1}_{\{\overline{\delta}_{\rho(k)}=1\}} \left\| \boldsymbol{B}_k - \overline{\boldsymbol{B}}_{\rho(k)} \right\|_1. \tag{4}$$

**Existence loss $\mathcal{L}_e$.** As we are matching two sets with possibly different number of elements, we synchronize the existence of each element appearing in the two sets. To enforce such a coherency, we minimize the cross-entropy for the existence probabilities between each pair of matched prediction and target bounding boxes:

$$\mathcal{L}_e(k) = \mathcal{H}(\delta_k, \overline{\delta}_{\rho(k)}). \tag{5}$$

Therefore the matching loss $\mathcal{L}_m$ of **an object** is defined as

$$\mathcal{L}_m = \sum_{k=1}^{NM} \lambda_s \mathcal{L}_s(k) + \lambda_v \mathcal{L}_v(k) + \lambda_e \mathcal{L}_e(k). \tag{6}$$

During optimization, we select the optimal permutation $\rho^*$ among all permutations $S_{NM}$ to compute $\mathcal{L}_m$ and apply gradient descent. Particularly, at each step, we search for $\rho^* = \arg\min_{\rho \in S_{NM}} \mathcal{L}_m$ using Hungarian algorithm [29]. In our experiments, we use $\lambda_s = 1$, $\lambda_v = 0.2$, and $\lambda_e = 0.8$ for all categories.

Here, we provide an intuitive understanding of our optimization process. First, our objective anchors one copy of an initialized shape to a nearby part on the object with similar scales, e.g., a leg of a

| Method | IoU ↑ | | | F-Score@0.01 ↑ | | | Method | mIoU ↑ | | |
|---|---|---|---|---|---|---|---|---|---|---|
| | Chair | Table | Airplane | Chair | Table | Airplane | | Chair | Table | Airplane |
| Shape2Prog [55] | 0.365 | 0.345 | N/A | 20.47 | 25.76 | N/A | Shape2Prog [55] | 0.549 | 0.743 | N/A |
| CubeSeg [64] | 0.315 | 0.238 | 0.360 | 25.56 | 30.38 | 42.53 | CubeSeg [64] | 0.737 | **0.895** | 0.688 |
| BSP-Net [9] | 0.541 | 0.421 | 0.472 | 42.94 | 36.18 | 46.91 | BSP-Net [9] | 0.669 | 0.859 | 0.721 |
| ProGRIP (ours) | **0.620** | **0.656** | **0.680** | **57.98** | **72.30** | **76.84** | ProGRIP (ours) | **0.752** | 0.857 | **0.757** |

| | |
|:---:|:---:|
| **ShapeNet Reconstruction** | **Unsupervised Part Segmentation** |

Table 1: **Performance on ShapeNet.** To quantitatively evaluate ProGRIP, we report the reconstruction quality using IoU and F-Score, and unsupervised part segmetnation performance using mIoU. For reconstructions, ProGRIP outperforms prior approaches by a margin. For unsupervised part segmentations, ProGRIP tops 2 out of 3 categories. The results of Shape2Prog [55] on airplanes are not available, as the domain specific language for airplanes is not defined in their original paper.

chair, due to the joint impact of $\mathcal{L}_s$ and $\mathcal{L}_v$. Then, as the gradient on scales will be propagated to other copies of the same shape, these copies will converge to leg shapes as well even if they are not matched to legs initially. In later rounds of optimization, these copies' poses will also converge to the nearest legs on the chair because of the higher weight on $\mathcal{L}_s$. In the end, ProGRIP learns the recurring leg shapes on a chair that are posed correctly.

## 4.2 Learning to Execute ProGRIP

After the optimization in Sec. 4.1, ProGRIP can predict the repeatable oriented bounding boxes of each shape element. We further train ProGRIP to refine the shape element within each box as posed implicit functions. As defined in Sec. 3.2, the outcome of executing ProGRIP is an object occupancy function $\mathcal{O}$. To optimize this function, we uniformly sample points $\boldsymbol{x}$ within the padded bounding box of an object and collect the ground truth occupancy for each points $\overline{\mathcal{O}}(\boldsymbol{x})$, where $1$ represents inside points and $0$ outside. We further augment the samples by sampling points near the surface and sampling the point cloud of the object. Note that we define the occupancy of on-surface points $\boldsymbol{x} \in \partial\overline{\mathcal{O}}$ as $\overline{\mathcal{O}}(\boldsymbol{x}) = 0.5$. The loss $\mathcal{L}_o$ for a set of $C$ samples from an object is averaged cross entropy:

$$\mathcal{L}_o = \frac{1}{C} \sum_{\boldsymbol{x}} \mathcal{H}(\mathcal{O}(\boldsymbol{x}), \overline{\mathcal{O}}(\boldsymbol{x})). \tag{7}$$

## 5 Experiments

We conduct all our experiments using the ShapeNet [6] dataset following ShapeNet Terms of Use. We quantitatively evaluate the performance of our model on the reconstruction (Sec. 5.1) and part segmentation (Sec. 5.2) tasks. Additionally, we dive deep into our method via investigating the impact of our unsupervised matching loss (Sec. 5.3). We also showcase *applications of ProGRIP* in shape editing and compact shape representations in our **supplementary material**, along with failure cases.

**Metrics**. We quantitatively evaluate our method on its reconstruction quality and segmentation accuracy. For reconstruction, we use two quantitative criteria: ① The volume-based Intersection-over-Union (IoU), where we uniformly sample points and evaluate the IoU of our predicted occupancies and the ground truth occupancies. Note that this metric is focused on the accuracy of the volume reconstruction. ② The surface-based F-Score, where we follow the protocol defined in Tatarchenko et al. [54] to compute the "percentage of surface that is correctly reconstructed". It is claimed to be the best metric to assess the fidelity of the reconstruction, with a concentration on the surface. For segmentation, we use the mean per-label Intersection-over-Union (mIoU) to measure the alignment of our predicted segmentation and the ground truth labels. As our segmentation task is unsupervised, to compute the mIoU during evaluation, we assign the ground truth label to each predicted label by a majority voting, i.e., picking the ground truth label with the highest correlation in the training split.

**Baselines**. We compare against representative state-of-the-art methods from relevant domains as discussed in Sec. 2. Specifically, we select CubeSeg [64] for primitive-based shape representation, Shape2Prog [55] for shape programs, and BSP-Net [9] for implicit or hybrid shape representations. Note that while LDIF [21] is also an implicit part-based representation, it strives for high fidelity

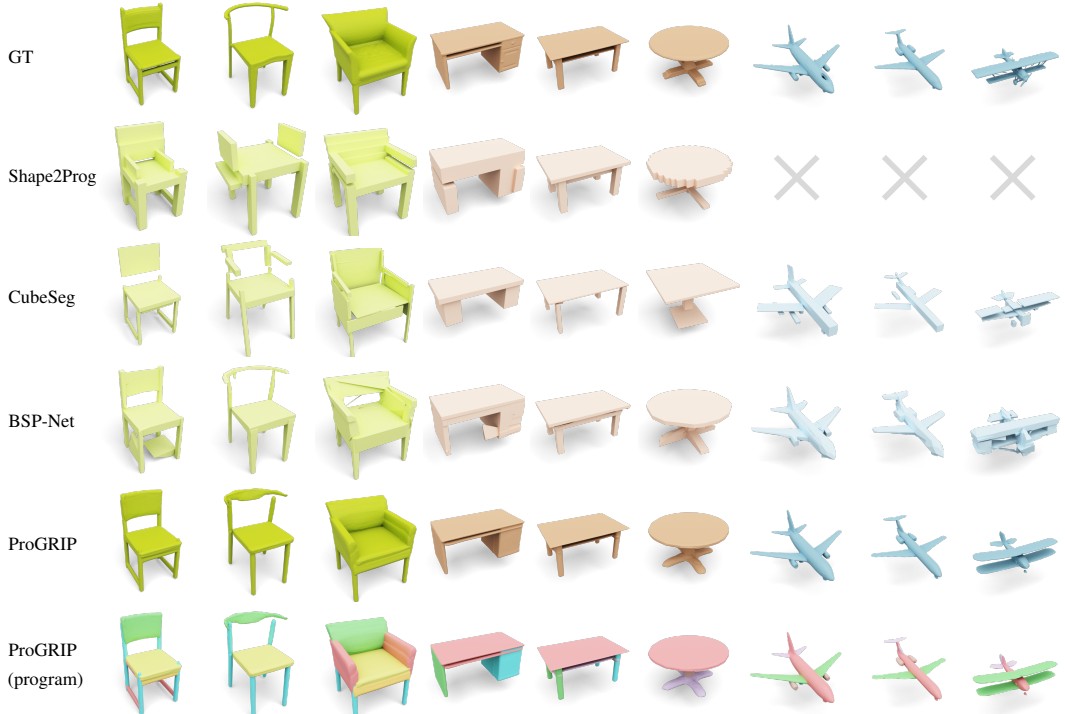

Figure 5: We qualitatively compare ProGRIP with state-of-the-art shape decomposition methods (including Shape2Prog [55], CubeSeg [64], and BSP-Net [9]) by rendering out their reconstructions. For ProGRIP, we present both plain reconstruction (the last second row) as well as per-part colored rendering (the last row). The ground truth mesh is shown in the first row. ProGRIP reconstructs shapes more accurately and smoothly. Note that the cylindrical parts on the chair back (col. 1 and col. 2) are reconstructed as different copies of the same shape as 4 chair legs for their geometric similarity. Meanwhile, since there are asymmetric table legs in ShapeNet (col. 4), our method discovers symmetries for left and right legs independently (col. 5).

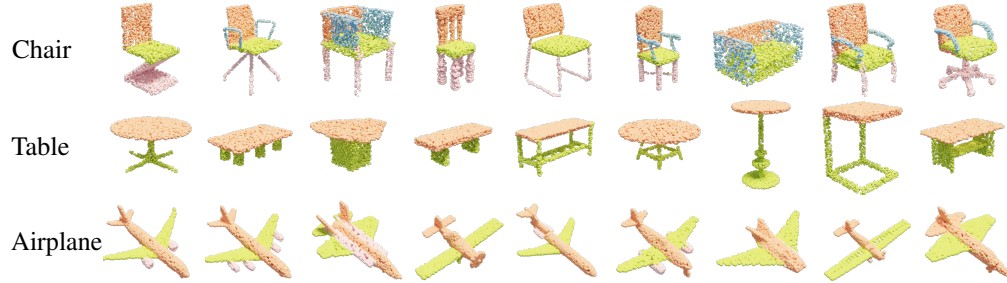

Figure 6: We visualize the unsupervised part segmentation results produced by our ProGRIP, which generalizes well across different object categories. Besides, ProGRIP is capable of segmenting fine-grained parts, such as engines and tails of aircrafts.

reconstruction rather than better structural understanding. We don't directly compare to it due to different encoding schemes, i.e. local feature encoding (LDIF) vs. global feature encoding (ProGRIP).

**Implementation details**. We use an auto-encoding architecture for our model. The encoding architecture is identical to Yang and Chen [64] and the programs are predicted by 2 levels of Transformers [58]. We first train our ProGRIP generator (Fig. 2) using the decomposition from Yang and Chen [64] and $\mathcal{L}_m$. Then we train the ProGRIP execution module (Fig. 3) as well as fine-tune the program generator by fitting occupancy predictions to ground truth occupancies of samples. Please see our supplementary materials for more implementation details.

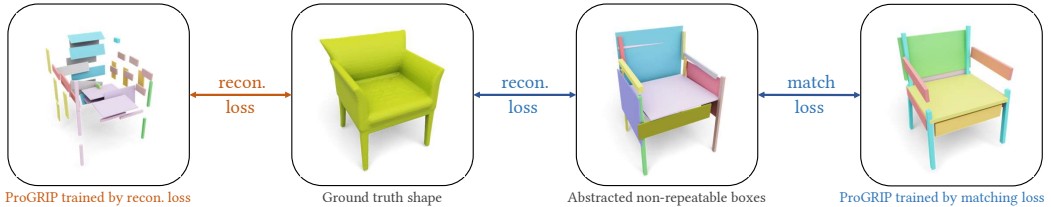



ProGRIP trained by recon. loss     Ground truth shape     Abstracted non-repeatable boxes     ProGRIP trained by matching loss



Figure 7: **Our Matching Loss $\mathcal{L}_m$ vs. Reconstruction Loss in Yang and Chen [64]**. We use either our matching loss $\mathcal{L}_m$ **(bottom)** or the reconstruction loss in Yang and Chen [64] **(top)** to train a ProGRIP and visualize their results on one test example. Note that matching loss uses the abstracted *non-repeatable* boxes that are unsupervised learned by reconstructing the ground truth shape (vertical arrow). Therefore, both losses are fully unsupervised. However, we observe clearly better reconstruction quality and more plausible part arrangement for the ProGRIP trained by $\mathcal{L}_m$ on bottom right than trained by reconstruction loss on top right.

## 5.1 Shape Reconstruction

As we extend independent non-repeatable primitive modeling to repeatable geometry modeling, we boost the efficiency of geometry learning — the same repeatable part can receive gradients from its copies at various local neighborhoods of the object. Additionally, we employ implicit shape representations as our neural implicit primitive to greatly enrich the capacity of our primitive shape space. We validate these two improvements on the point cloud-to-mesh 3D reconstruction task. In this task, each model takes as input a point cloud with 4096 points sampled from the object surface and strives for reconstructing the object surface with high fidelity. We compare against baseline methods quantitatively in Tab. 1, Reconstruction and display a few qualitative visualizations in Fig. 5. We find that our method consistently outperforms other primitive-based baselines in reconstruction quality. Meanwhile, our predicted repeatable parts are reasonably posed to fit local geometry with similar or identical shapes, such as the four legs of a chair.

## 5.2 Unsupervised Part Segmentation

We further quantitatively study the primitive assignment in our shape programs on the unsupervised part segmentation task. In this task, we aim to learn to segment each object into semantic parts without any segmentation supervision. Specifically, we provide the point cloud of an object to each method and ask for per-point semantic part segmentation labels. Each method is trained on reconstruction tasks only and we use their part decomposition at test time to evaluate the segmentation quality. In Tab. 1, Unsupervised Part Segmentation, we show the per-label mIoU of different primitive-based methods and find that our ProGRIP achieves competitive performance in comparison to state-of-the-art unsupervised methods, such as CubeSeg and BSP-Net; it even outperforms Shape2Prog, which uses synthetic semantic structure ground truth for pretraining.

## 5.3 Analysis: Impact of Matching Loss

We present the matching loss $\mathcal{L}_m$ in Sec. 4.1 as a better fit for identifying and learning repeatable parts than the commonly used reconstruction loss. Here we provide an experimental study to validate our design choice. We visualize the results in Fig. 7. In this study, we train the same ProGRIP model using either our matching loss $\mathcal{L}_m$ on abstracted non-repeatable boxes or the reconstruction loss directly from the ground-truth shape as defined in Yang and Chen [64]. We find that ProGRIP learned from the matching loss produces both decent quality of reconstruction and meaningful repeatable parts arrangement. This can be observed in the symmetric arms of the chair that is fitted by two copies of the same repeatable part and also in the four repeated legs. In contrast, the part allocation of ProGRIP learned directly from the reconstruction loss appears to be of poor quality elucidating the difficulty of learning part decomposition directly from reconstruction. The reconstruction itself is also crippled by the unwise part assignment as each repeatable part is receiving conflicting gradients.

# 6  Conclusion

We propose to represent objects as shape programs with repeatable implicit parts, or ProGRIP, a structured high fidelity shape representation that can be learned without structure annotations. While prior works on shape programs suffer from poor fidelity due to the limited representation capacity of their coarse shape primitives, we significantly augment the reconstruction quality by introducing implicit functions to implement the predicted parts. Additionally, we design an unsupervised training objective to facilitate the learning of ProGRIP on *any* object category, largely enhancing the scalability of shape programs compared to prior works that only work with few categories with ground truth semantic structures. On the shape reconstruction task, our ProGRIP champions on all categories among structured representations. It also achieves top-tier part segmentation accuracy and is shown to be an accurate yet highly compact representation capturing repetitions and symmetries of object parts, enabling efficient shape editing.

## Discussion

**Broader Impact**. This paper studies how to capture the regularities (e.g., *geometry* and *structure* priors) of shapes by programs. We believe it will be beneficial to the field of shape representation, and it can potentially be used by practitioners to edit existing CAD models or build new ones from scratch. This algorithm we build is unsupervised, which may have the beneficial effects of being more cost effective and reducing biased introduced by human annotations. Meanwhile, we are aware that the whole system is not completely bias free: the domain specific language designed may still be biased. At this point, we do not foresee any direct ethical issues and concerns that may rise with our system, but we urge the users to be mindful about the potential bias.

**Acknowledgments**. We'd like to thank Pratul Srinivasan for suggesting the compactness application of ProGRIP, Kaizhi Yang for kindly open-sourcing the code of CubeSeg [64], the Occupancy Networks [37] team for releasing their pre-processed occupancy samples, and the Stanford Artificial Intelligence Laboratory (SAIL) staff for maintaining and supporting our computational infrastructures. This work is in part supported by the Stanford Institute for Human-Centered Artificial Intelligence (HAI), Center for Integrated Facility Engineering (CIFE), Toyota Research Institute (TRI), NSF RI #2211258, NSF CCRI #2120095, ONR MURI N00014-22-1-2740, and Adobe, Amazon, Analog, Autodesk, IBM, JPMC, Meta, Salesforce, and Samsung. BD is funded by a Meta Research PhD Fellowship. SK is in part supported by the Brown Institute for Media Innovation.

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
