# Unsupervised Learning of Shape Programs with Repeatable Implicit Parts
## Supplementary Materials

**Boyang Deng**[1,*]    **Sumith Kulal**[1,*]    **Zhengyang Dong**[1]    **Congyue Deng**[1]

**Yonglong Tian**[2]    **Jiajun Wu**[1]

[1]Stanford University,   [2]MIT,   *equal contributions

This document contains additional materials for our proposed method, ProGRIP, including additional details of our implementation (Sec. A), two demonstrative applications in semantic shape editing (Sec. B) and shape compactness (Sec. C), our study on using *repeatable parts* instead of *posed parts* for segmentation (Sec. D), a comparison on reconstruction quality between our box abstractions and our posed implicit functions (Sec. E), a comparison on reconstruction quality between repeatable parts and nonrepeatable parts both using implicit shape representations (Sec. F), more baselines (Sec. G), more qualitative visualizations (Sec. H), and some representative failure cases of our method (Sec. I). Please visit our project webpage, **progrip-project.github.io**, for additional visualizations.

## A    Additional Implementation Details

**Architecture**. We use an auto-encoding architecture for generating ProGRIP programs. The encoding network uses an identical architecture as in [12], which takes in a pointcloud of 4096 points and predicts a 512-d feature vector as the *global feature* of the input. Then this global feature vector is feed into a *geometry transformer* to predict as set of repeatable parts, specified by their scale $s_i$ and part latent $z_i$. The geometry transformer is constructed by stacking 4 self-attention blocks [11], each with 8 attention head, 256-d hidden features, and 2048-d feed-forward features. Note that this architecture is the same as the transformer used in [1]. Further, to predict the occurrences of each repeatable part, we use another *pose transformer* with the same architecture as the geometry transformer to transform the part latent $z_i$ to a set of posed occurrences parameterised by the translation $t_{i,j}$, rotation $R_{i,j}$, and existence probability $\delta_{i,j}$. At program generation, we set the maximum number of repeatable parts $N = 10$ and the maximum number of occurrences for each repeatable part $M = 6$. So far, the architecture is already capable of predicting a ProGRIP composed of posed parts $\{(s_i, z_i, t_{i,j}, R_{i,j}, \delta_{i,j})\}$. To further model the fine geometric details of the object, we execute each posed part as a pose implicit function that can answer occupancy queries for any point $x$. Particularly, given $x$, we first inverse transform it by $x' = \left(R_{i,j}^{-1}\left(x - t_{i,j}\right)/s_i\right)$. Then we concatenate it with $z_i$ and feed it into an implicit part decoder $\mathcal{P}$. We use the same architecture for $\mathcal{P}$ as in [6] except that we use 50-d latent features instead of 512-d for efficiency purpose. The output of $\mathcal{P}$ is multiplied by the binary existence $\hat{\delta}_{i,j}$ to get the occupancy of $x$ regarding posed part $(i, j)$, $o_{i,j}(x)$. The object occupancy $\mathcal{O}$ is the maximum of all $o_{i,j}$.

**Optimization**. We train our model in 2 stages. At the first stage, we optimize the parameters for generating ProGRIP. Specifically, we use stochastic gradient descent to minimize $\mathcal{L}_m$. The ground truth *non-repeatable* box abstraction is obtained by running the officially released pretrained models from [12]. We use an AdamW [5] optimizer with a learning rate of 0.0001 and a batch size of 32. We train the model for 75 epochs which is ~2 hours on an Nvidia Titan RTX. At the second stage, to train the implicit functions, we use the sample points preprocessed by [6] and [3]. In particular, for each

36th Conference on Neural Information Processing Systems (NeurIPS 2022).

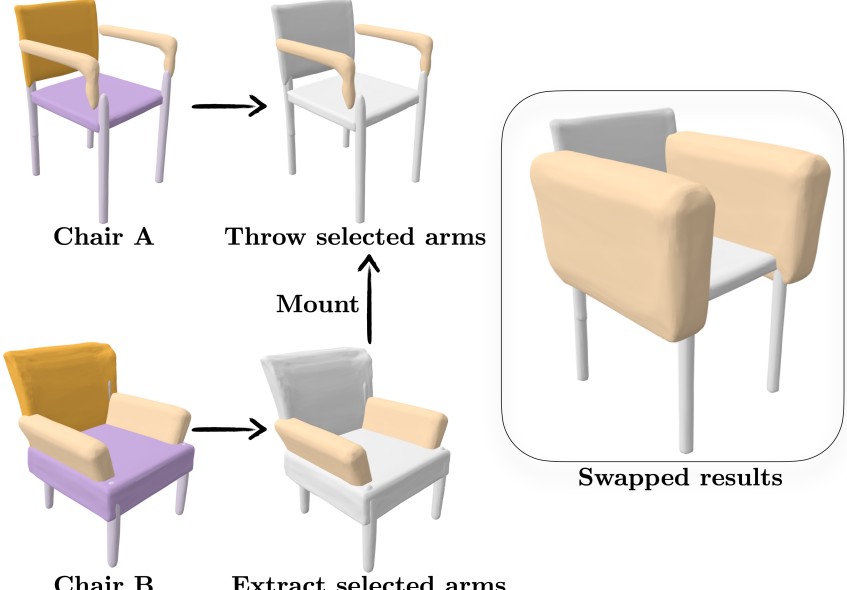

Chair A     Throw selected arms

Mount

Swapped results

Chair B     Extract selected arms

Figure 1: We show an example editing application of ProGRIP where we swap semantic parts. First, we take 2 chairs reconstructed by ProGRIP with semantic part decomposition (**left**). Then, one can select one or multiple semantic parts on the chairs that they want to swap. In this case, we select chair arms on both chairs (**middle**). The swapping can be done as simple as swapping the shape latent of those arms (**right**). Note that while we only swap *one* shape latent, the shape changing is automatically propagated to both arms and the scale and pose are aligned properly to the new chair thanks to the disentangled shape and pose modeling in ProGRIP.

shape within a batch, we use 1300 points uniformly sampled from the bounding box of the object, 200 points sampled near the object surface, and 100 points sampled on the object surface. We use an Adam optimizer for parameters within the implicit part decoder $\mathcal{P}$ and an AdamW optimizer for all the other parameters. The learning rate of both optimizers is set to 0.0001 and the batch size is 16.

**Evaluation**. We validate ProGRIP on 2 tasks, reconstruction and unsupervised part segmentation. For reconstruction, we measure the quality by computing the IoU and F-Scores. We use the same $100,000$ volume samples with ground truth occupancy as [6] to compute the IoU. To compute the F-Score, we first extract the meshes for all objects. Then we sample $100,000$ points from both the predicted mesh and the ground truth mesh and calculate the F-Score at a threshold $\tau = 0.01$. As for segmentation, we follow prior works [2, 12] to use the ShapeNet-Part [13] part annotations and merge the labels for *leg* and *support* of tables as 1 semantic label. We further merge the *tail* and *body* labels for we found inconsistent labeling within these 2 parts.

While we believe that we've provided all the implementation details to reproduce our results, **upon publication, we will release our code for all the results reported** to further assist future research built on our work.

## B   Application: Semantic Shape Editing

After training, our ProGRIP can decompose shapes into recurring semantics structures. This enables more convenient shape editing as one can directly operate at the semantic level such as both arms of a chair rather than the element level such as each arm alone. In Fig. 1, we demonstration one demo editing where we swap the arms of 2 different chairs. With the help of ProGRIP, this processing can be as simple as swapping the shape latent code. Note that the scale, orientation, and location of the swapped arms are automatically aligned to the new base chair thanks to the 2-level modeling of shape and pose in ProGRIP. We present additional editing examples in Fig. 2.

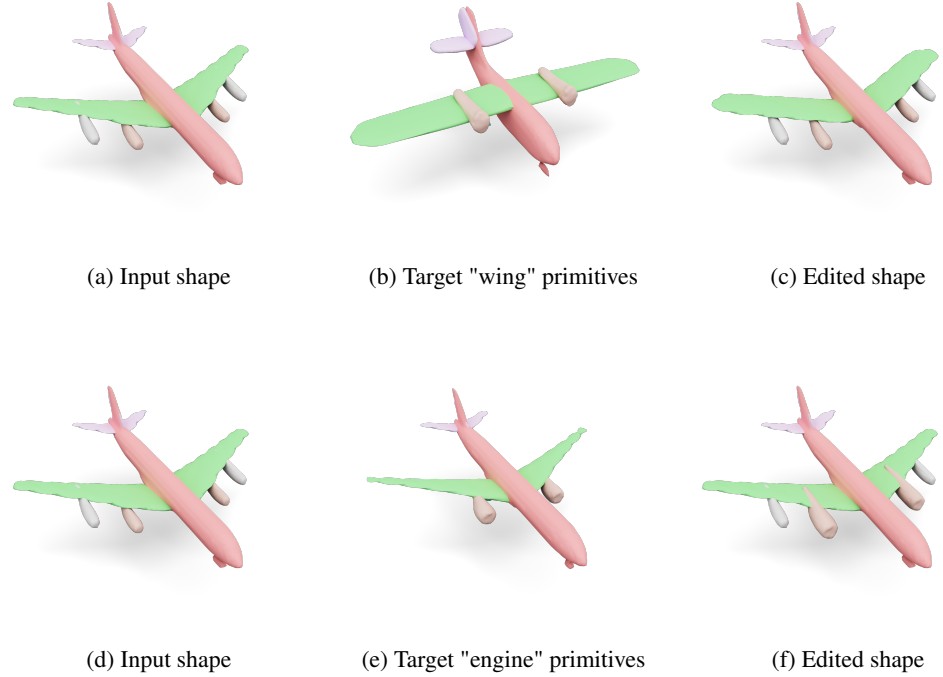

(a) Input shape  (b) Target "wing" primitives  (c) Edited shape

(d) Input shape  (e) Target "engine" primitives  (f) Edited shape

Figure 2: **More editing examples.** We present some more editing examples enabled by our ProGRIP representation. Given an input shape (left), we swap latent-codes for a target primitive from a different shape (middle) to generate the edited output (right).

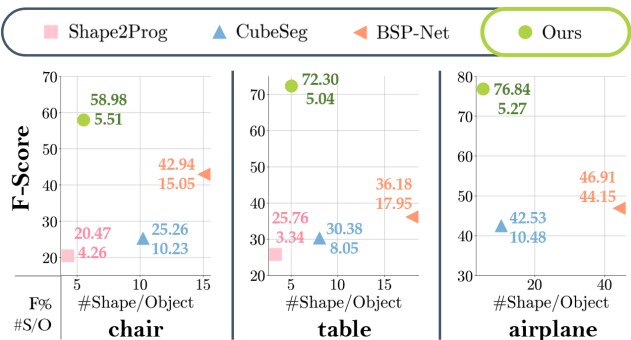

Figure 3: We compare ProGRIP with Shape2Prog [10], CubeSeg [12], and BSP-Net [2] on **Shape Compactness**, which requires the model to be concise and expressive simultaneously. ProGRIP possesses significantly better expressivity (measured by F-score) while achieving similar compactness as Shape2Prog.

## C  Application: Shape Compactness

Along the development of machine intelligence especially machine learning algorithms, a clear connection has been made between compactness and learning [8, 9]. An alternative perspective of understanding machine learning is essentially a process of finding the most succinct representation of the information in data. In primitive-based shape representations, we pursue a structured intelligent understanding of object shapes. Hence, being able to represent shapes concisely can effectively demonstrate a capability of understanding. To this end, we evaluate our method on a simple shape compactness application. In this task, we ask each primitive-based method to reconstruct the same set of objects (the test split of ShapeNet) and compute the average number of shapes used to represent

| Method | Instance Segmentation | | | Semantic Segmentation | | |
|---|---|---|---|---|---|---|
| | Chair | Table | Airplane | Chair | Table | Airplane |
| Shape2Prog [10] | 0.549 | 0.743 | N/A | 0.493 | 0.742 | N/A |
| ProGRIP (ours) | **0.752** | **0.857** | **0.757** | **0.684** | **0.850** | **0.755** |

Table 1: **Unsupervised Part Segmentation by Repeatable Parts.** Here we compare the mIoU of part segmentation (higher is better) on either the instance segmentation setup or the semantic segmentation setup. Note that the semantic segmentation setup makes the task harder as one has to correctly associate different instances of the same semantic part, e.g. four legs of a chair. We compare against only Shape2Prog [10] here as other methods such as CubeSeg [12] and BSP-Net [2] is not capable of modeling any associations among parts.

each object. Note that we focus on the compactness of structure abstraction in this study so that only the shape count is considered as the compactness measure. This measure is therefore agnostic to the actual shape representation used. We show the compactness of representation (in # Shapes / Object) in Fig. 3 as well as the reconstruction quality (in F-score) as a reference. Our method is able to achieve the second least average shape counts. Particularly, our shape representation is about $9\times$ more compact than the most complex BSP-Net. Note that Shape2Prog uses structure supervised pretraining, where the compression heuristics is already manually baked in the ground truth structures. Since ProGRIP's shapes are decoded from a latent feature while methods such as CubeSeg [12] and Shape2Prog [10] only use boxes or voxel grids, we speculate the high fidelity largely comes from the the implicit shape representation we introduce to our shape programs. Please refer to Tab. 2 and Tab. 3 for extended comparison when we align the shape representation across methods.

## D  Part Segmentation by Repeatable Parts

In this section, we extend our evaluation of unsupervised part segmentation from instance segmentation to semantic segmentation. As most prior works including CubeSeg [12] and BSP-Net [2] only support non-repeatable parts and do not model any relationships among parts, they are unable to perform the semantic segmentation task as different instances, e.g. four legs of a chair, of a semantic class, e.g. chair leg, have to be grouped together. Therefore, we only compare all 4 methods on the instance segmentation task. In this task setup, despite that our ProGRIP would model 4 legs of a chair as 4 occurrences of the same repeatable part with different poses, we treat them as different part labels before assigning semantic labels to them by voting. To further assess the learned geometric regularities such as symmetry and shape recurrences, we use the repeatable parts from ProGRIP and Shape2Prog for semantic part segmentation in Tab. 1. Under the semantic segmentation evaluation setup, we strictly assign the same semantic labels for all occurrences of the same repeatable part, e.g. 4 legs of a chair. We find that while Shape2Prog is pretrained by ground truth shape regularities, our fully unsupervised ProGRIP consistently outperforms it on both instance segmentation and semantic segmentation.

## E  Box Abstraction *vs*. Implicit Functions

Different from using simple predefined parametric primitives [12] or coarse voxel grids [10] in prior works, we propose to use posed implicit functions to execute our shape programs. Implicit representations have demonstrated excellent capability of modeling complicated signals [6, 7]. Consequently, thanks to the introduction of posed implicit function to our framework, we also observe notable boosts of reconstruction quality. In Tab. 2, we show an ablation study where we use either the box abstractions defined by the $(\boldsymbol{s}_i, \boldsymbol{t}_{i,j}, \boldsymbol{R}_{i,j}, \delta_{i,j})$ tuples in our ProGRIP or the posed implicit functions for shape reconstruction. We find that from box to implicit the performance leap can be as much as $\mathbf{2}.\mathbf{4}\times$ better IoU for the Table class. More surprisingly, we find that our box abstractions are also more geometrically accurate than the CubeSeg baseline, although it is trained on the boxes from CubeSeg. We claim that the improvement is mostly likely to be the outcome of a more structured formulation, especially with among-part relationships considered. For example, while CubeSeg learns 4 legs of a chair independently, our ProGRIP optimizes the shape parameters, $\boldsymbol{s}_i$ and

| Method | IoU ↑ | | | F-Score@$0.01$ ↑ | | |
|---|---|---|---|---|---|---|
| | Chair | Table | Airplane | Chair | Table | Airplane |
| CubeSeg [12] | 0.315 | 0.238 | 0.360 | 25.56 | 30.38 | 42.53 |
| ProGRIP (box) | 0.423 | 0.276 | 0.376 | 35.70 | 36.57 | 48.81 |
| ProGRIP (implicit) | **0.620** | **0.656** | **0.680** | **57.98** | **72.30** | **76.84** |

Table 2: **Box *vs*. Implicits in Reconstruction.** We compare the reconstruction quality of our model, ProGRIP, using either the box abstractions (box) as specified by $(s_i, t_{i,j}, R_{i,j}, \delta_{i,j})$ or the high fidelity posed implicit functions (implicit). We find clear boost in reconstruction quality from box to implicit. As a reference, we also report the reconstruction performance o non-repeatable box abstractions from CubeSeg [12]. We fit our repeatable box abstraction to CubeSeg's boxes during the learning of program generation. Yet, due to our program formulation, our model learns more regularities than CubeSeg, e.g. we require legs of a chair to be identical in shape and likely to be symmetric as well. Hence, our box abstractions achieve higher reconstruction qualities in both IoU and F-Score than CubeSeg.

| Method | CubeSeg [12] + Implicit Parts | | | ProGRIP | | |
|---|---|---|---|---|---|---|
| | Chair | Table | Airplane | Chair | Table | Airplane |
| IoU ↑ | 0.549 | 0.599 | 0.616 | **0.620** | **0.656** | **0.680** |

Table 3: **Repeatable Implicit Parts *vs*. Nonrepeatable Implicit Parts.** We extend CubeSeg [12] to representing each part predicted by CubeSeg as neural implicit functions. The training of such implicit parts is identical to ProGRIP. While both CubeSeg and ProGRIP use the same data for training, ProGRIP differs from CubeSeg in that it enables the modeling of repeatable geometric structure, thanks to our unsupervised training strategy. Results show that this greatly help the reconstruction quality.

$z_i$, of these instances simultaneously, resulting in stronger gradients and faster convergence. The self-attention architecture with our geometry and pose transformers can also help learn regularities such as symmetric poses.

## F   Repeatable Implicit Parts *vs.* Nonrepeatable Implicit Parts

To further investigate the impact of having repeatable parts based on neural implicit functions, we compare our ProGRIP to an extended baseline where we plug implicit shape representations to the shape programs generated by CubeSeg [12]. Specifically, we use an identical training of implicit parts as Sec. 4.2 of the main paper for CubeSeg. The results are shown in Tab. 3. We find that by having recurring geometric structured accounted for in ProGRIP, enabled by our unsupervised training strategy, the reconstruction quality are consistently improved across all categories.

## G   More Baselines

While we select representative baselines from all relevant domains to compare ProGRIP against, we further extend our study with additional relatively more loosely related baselines (Tab. 4). Specifically, we compare with OccNet [6] which shares the spirit of using implicit shape representation with ProGRIP but strives for a different goal of only reconstructing each shape as a whole with high quality. In other words, any structure such as part-to-whole relationships are ignored in OccNet. We speculate that OccNet achieves higher reconstruction quality as a results of no constraints put on its shape modeling. In contrary, ProGRIP has to use strictly identical shapes to reconstruct multiple elements that may have minor geometric variations. Exploring loosening this hard constraint but still keeping the similarity relationship for better quality would naturally be an exciting future direction. Meanwhile, we also compare ProGRIP to SIF [3] that also uses a part-based representation. However, ProGRIP is capable of representing more complicated structure, i.e. recurring geometry, while SIF treats each part independently. We find in Tab. 4 that this more advanced structure modeling helps

| Method | IoU ↑ | | | F-Score@0.01 ↑ | | |
|---|---|---|---|---|---|---|
| | Chair | Table | Airplane | Chair | Table | Airplane |
| OccNet [6] | 0.739 | 0.756 | 0.770 | 77.20 | 84.90 | 87.80 |
| SIF [3] | 0.503 | 0.485 | 0.660 | 48.81 | 60.02 | 82.07 |
| ProGRIP | 0.620 | 0.656 | 0.680 | 57.98 | 72.30 | 76.84 |

Table 4: **More Baselines.** In addition to the comparison in our main paper, we provide more baselines as a reference for ProGRIP's performance on reconstruction. Here we compare ProGRIP against OccNet [6] and SIF [3]. Note that while OccNet achieves the highest reconstruction quality as its sole goal, it doesn't predict any structure such as geometric or semantic parts of objects. Hence, it stands closer to the shape reconstruction domain instead of shape understanding. SIF is closer to our method but with nonrepeatable parts in a different representation.

improve the reconstruction quality. Note that we didn't compare with LDIF [4] for that LDIF is using a more complex local feature encoding that is vital to their reconstruction quality. While it's not within the scope our this work, we enthusiastically encourage the community to carry on investigating incorporating local feature encoding to ProGRIP.

## H   More Qualitative Results

In additional to the qualitative examples shown in the main paper, we provide more renderings of our reconstruction results in Fig. 5 and Fig. 6. As a comparison, we also show results from other state-of-the-art baselines (Shape2Prog [10], CubeSeg [12], and BSP-Net [2]). We find that reconstructions from our ProGRIP consistently preserve more geometric details than baseline methods. Moreover, ProGRIP successfully learned geometric regularities of object shapes, such as recurring geometry and symmetry (see the "program" rows).

In Fig. 7, we further display the unsupervised part segmentation results from our ProGRIP, as well as Shape2Prog [10], CubeSeg [12], and BSP-Net [2]. We observe that while baseline methods occasionally annotate asymmetric semantic labels to symmetric semantic parts (e.g. arms of chairs and engines of airplanes), ProGRIP reliably maintains the symmetric structures in semantic labeling, thanks to the formulation of repeatable parts.

## I   Failure Cases

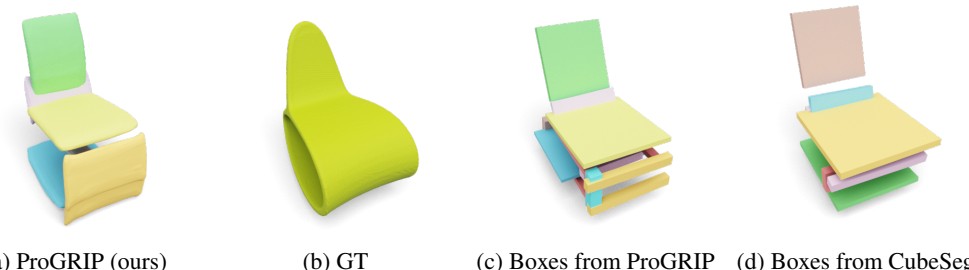

| (a) ProGRIP (ours) | (b) GT | (c) Boxes from ProGRIP | (d) Boxes from CubeSeg |

Figure 4: **Failure Case Renders.** We display in this figure a representative failure case of ProGRIP. The ground truth shape (b) is extremely irregular. Hence, it's difficult for ProGRIP to capture the geometric structure. As a result, the reconstruction is imperfect. We analyze the source of errors and find that the box-based reconstruction from CubeSeg (d) has relatively low quality. As we train our program by matching our repeatable boxes (c) to CubeSeg's non-repeatable boxes, the mistakes are inherited and further passed on to our posed implicit functions (a).

While ProGRIP is able to learn the structures and repetitions of most shapes in a category, it fails to reconstruct objects that are extremely irregular. Especially difficult are the cases when the box abstractions from CubeSeg are incorrect. In Fig. 4 we show a representative example of such failure

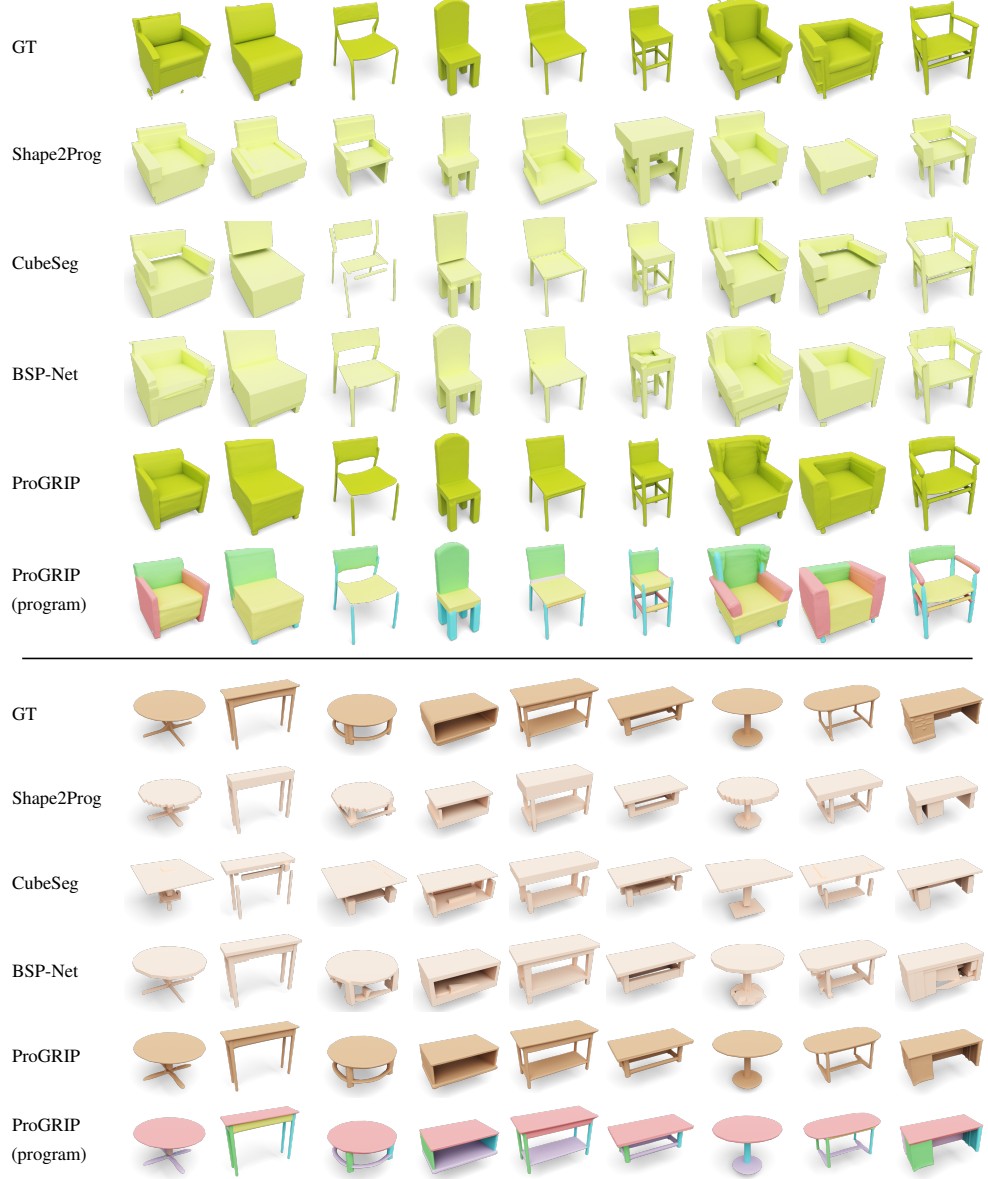

Figure 5: Additional reconstruction results of our ProGRIP and other state-of-the-art shape decomposition methods (Shape2Prog [10], CubeSeg [12], and BSP-Net [2]) on chair and table classes. The first row of each group shows the ground truth meshes.

cases. When the non-repeatable box-based reconstruction from CubeSeg is wrong, it misleads the optimization for our program generation. Consequently, the repeatable box abstractions from our predicted program inherit the mistakes. Even though during fine-tuning the posed implicit functions learn to partially amend the issues, the overall reconstruction quality still suffers from legacy errors. We believe that exploring better alternatives to CubeSeg and jointly learn program generation and shape reconstruction are promising directions to mitigate such failures.

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

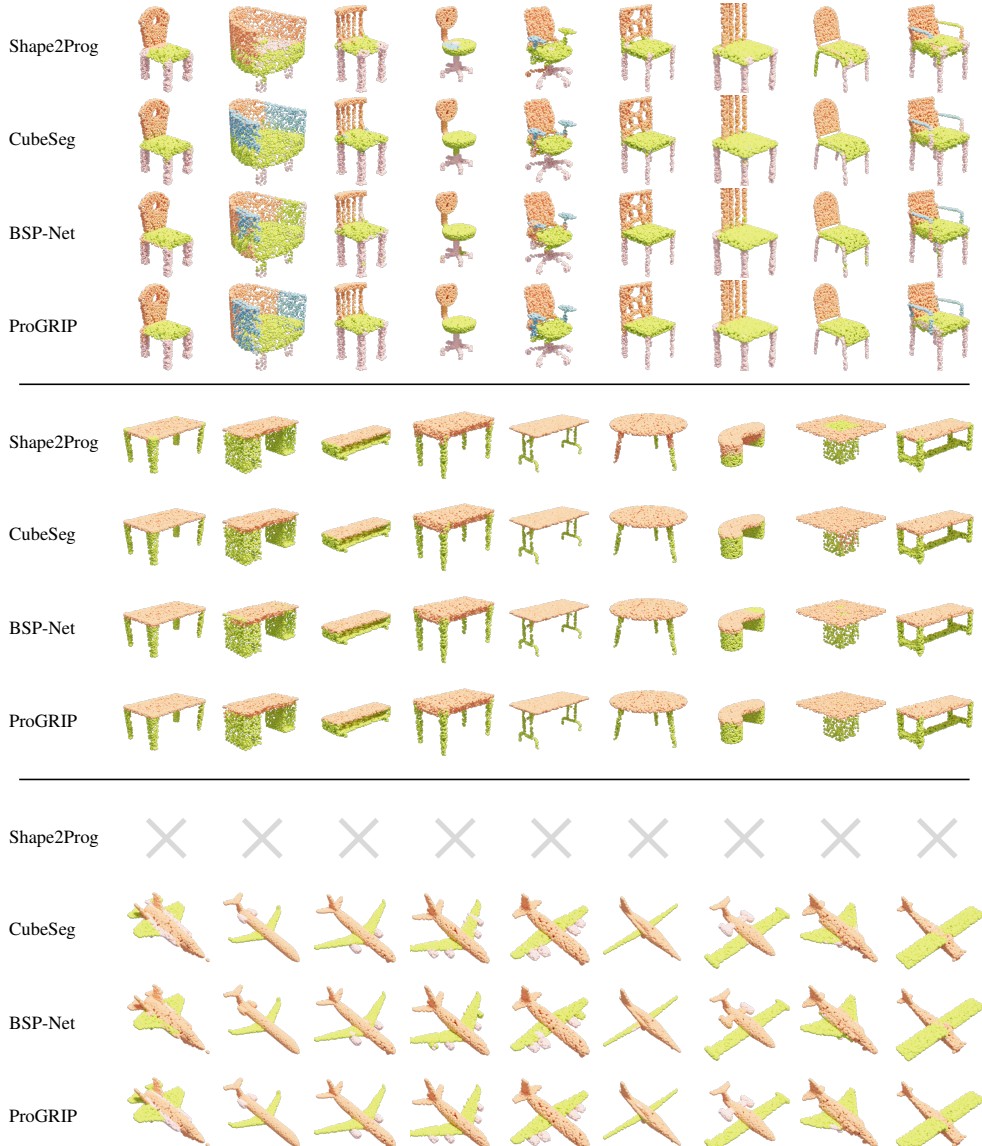

Figure 7: Comparison of ProGRIP with Shape2Prog [10], CubeSeg [12], and BSP-Net [2] on unsupervised pointcloud co-segmentation. **Top:** the chair class. Notice column 2 armrests, column 4 seat (Shape2Prog), column 5 armrests, column 8 legs (Shape2Prog & BSP-Net), and column 9 armrests. **Middle:** the table class. Notice column 2 top (BSP-Net), column 5 left leg (CubeSeg and BSP-Net), column 6 legs (Shape2Prog), column 7 & 8 top. **Bottom:** the airplane class. Notice column 1, 4, 5 jets (BSP-Net). Shape2Prog results are missing as it can only be trained with synthetic ground truth programs which are only available for chair and table.