# OpenReview forum: "Unsupervised Learning of Shape Programs with Repeatable Implicit Parts"
_NeurIPS.cc/2022/Conference — NeurIPS 2022 Accept_

### Official Review · Reviewer_SPGB · 2022-07-10

**Rating:** 6
**Confidence:** 4
**Soundness:** 3 good
**Presentation:** 3 good
**Contribution:** 3 good

**Summary:**

The paper proposes a new method for geometric and structure reconstruction of composite 3D shapes. The main contribution consist of automatically detecting and encoding the repeated parts. This leads to a relevant contribution in comparison with existing work. Moreover, the single parts are accurately reconstracted using neural implicit function. The proposed approach is fully unsupervised. An exhoustive experimental section shown the improvement of the proposed approach in comparison with other methods for both shape recontruction and shape segmentation.

**Questions:**

-A better explanation on the relation between the proposed approach and [55] is required to understand the impact of the proposed method (i.e., what happen if the method [55] fails).
-What is the computational complexity of the work in comparison with other methods?
-Authors emphasized that they proposed a shape program approach but it seems that  ‘program’ part is not considered in the methodological section and results. Authors should clarify why they consider their work as ‘program’-like (or clarify better the meaning of shape program).

Minor points:
Is it possible that scale s_i change for the same part? (i.e., two reapeted parts with different scales).
It is not clear the meaning of NM on line 193. Authors wrote that primitives are estimated using [55], but NM is predicted by ProGRIP, please clarify.


**Limitations:**

Limitations are not discussed. For instance authors should declare their strong dependency from [55].

**Strengths And Weaknesses:**

*Strengths*
-the paper focuses on a relevant problem that is the automatic encoding of composite objects. The capability of detecting repeatable parts is very important bringing a significative step toward the semantic understanding of objects and scenes.
-the use of deep implicit function enables the method to reconstruct more accurate details of the observed object improving the geometric representation of the observed parts  (differently than other method that capture only a coarse approximation of shape such as 3D boxes).
-the proposed composite encoding pipeline is well designed by properly combine existing methods with novel parts.
-  experiments are well organized and shown promising results in comparison with other methods.

*Weaknesses*
-It seems that the proposed work strongly believes on the results of [55]. In this sense the proposed work can be considered as an extension or a refinement of [55]. Authors should better explain this point.
-The proposed approach seems computational demanding, authors should discuss and evaluate the computational complexity of their work.
-The learning part is not fully novel since the main parts are based on alredy available neural architectures (for box estimation [55], and implicit function estimation [33, 38, 7]).

---

> ### Author Response · Authors · 2022-08-02
> **Response to Reviewer SPGB, thank you for the feedback!**
>
> - **Relation to CubeSeg [1]**:
>     - *Similarity*: as pointed out by the reviewer, ProGRIP relies on the box abstraction of shapes from CubeSeg [1] as a starting point for learning geometric self-similarity, i.e. repeatable parts. Therefore, it’s possible that some mistakes made by CubeSeg [1] would be inherited by ProGRIP. We discuss this in detail in Sec. I of our original supplementary materials and listed relaxing this dependency as an open future research direction.
>      - *Difference*: as we discussed in L109-113 (original main paper, or L110-114 in updated main paper) CubeSeg [1] uses non-repeatable parts and is limited in fidelity due to its box abstraction. In contrast, ProGRIP enriches the modeling capacity with repeatable parts and implicit shape representation. This is shown in significant improvement in reconstruction (Tab. 1). We also conduct an ablation study to extend CubeSeg with implicit shape representation and still show clear performance gains due to the use of repeatable parts (Tab. 3 in original supplementary materials).
>
> - **Computational complexity**: We show below the statistics on the computational complexity for ProGRIP. As a reference, we also show the same statistics for BSP-Net:
>     * | Method     | BSP-Net | ProGRIP |
> | ----------- | ----------- | ----------- |
> | Training |  ~7d10h  | ~2d 1 hour  |
> | Test   | 1.27s/mesh  | 1.35s/mesh  |
>      * From the statistics, we can see that ProGRIP is faster than BSP-Net during the training stage and comparable to BSP-Net on test time inference.
>
> - **Novelty in the learning part**: Please note that while ProGRIP inherits the **non-repeatable** box estimation from CubeSeg [1] and uses similar losses for implicit function training as [2, 3, 4], the core to our work is the learning of *repeatable geometric structure* as we outlined in Sec. 4.1. This is a novel training strategy and is critical to the success of our shape modeling (as shown in Fig. 7).
>
> - **Why is ProGRIP called a shape program?**: We follow standard terminology from the literature to classify ProGRIP as a shape program. In particular, we’d like to remark on the connection of our representation and the program in Shape2Prog [5] where the major difference is that our program is unordered while Shape2Prog’s program is sequential. For instance, all the occurrences of a repeatable part can be regarded as a `for loop drawing` in Shape2Prog where the `drawing` commands are executed in parallel.
>
> - **Varying the scale s_i for different posed parts from the same repeatable part?**: This is indeed an interesting idea. Currently, our formulation requires different posed occurrences of the same repeatable part to have identical shapes. One can relax such constraints by using a loss during the implicit shape training stage to encourage the shapes to be as close as possible instead. We list this as a good future extension of our work.
>
> - **Clarify NM in L193**: Please note in Sec. 3.1. (L141 - L 149) we formally define `N` as the number of repeatable parts and `M` as the number of posed occurrences for each part. `NM` is the product of these two values.
>
> - **Discussion on limitations**: Please see Sec. I in the original supplementary materials where we discussed our limitations and particularly visualized one example where our model is misled by the mistake in CubeSeg [1] (Fig. 3 in original supplementary or Fig 4. in updated supplementary).
>
> [1] Kaizhi Yang and Xuejin Chen. Unsupervised learning for cuboid shape abstraction via joint segmentation from point clouds. In SIGGRAPH, 2021.
> [2] Lars Mescheder, Michael Oechsle, Michael Niemeyer, Sebastian Nowozin, and Andreas Geiger. Occupancy networks: Learning 3d reconstruction in function space. In CVPR, 2019
> [3] Jeong Joon Park, Peter Florence, Julian Straub, Richard Newcombe, and Steven Lovegrove. Deepsdf: Learning continuous signed distance functions for shape representation. In CVPR, 2019.
> [4] Tao Chen, Saurabh Gupta, and Abhinav Gupta. Learning Exploration Policies for Navigation. In ICLR, 2019.
> [5] Yonglong Tian, Andrew Luo, Xingyuan Sun, Kevin Ellis, William T. Freeman, Joshua B. Tenenbaum, and Jiajun Wu. Learning to Infer and Execute 3D Shape Programs. In ICLR, 2019.

---

### Official Review · Reviewer_JZ3w · 2022-07-11

**Rating:** 5
**Confidence:** 4
**Soundness:** 3 good
**Presentation:** 3 good
**Contribution:** 2 fair

**Summary:**

The authors propose a method to reconstruct part-based shape models from point clouds, where each part has a local neural representation of the part's geometry. Additionally, multiple instances of the same geometry can be re-used by different parts, thereby explicitly modelling translational and rotational symmetries. The point cloud is encoded into a global feature vector, that is then used in two transformer-based steps to first generate part geometries and then translated and rotated instances of the part geometries. The method needs a part decomposition of the training shapes as supervision, although this decomposition can be obtained using existing unsupervised methods. A novel loss allows for efficient training of parts without requiring explicit correspondences with the ground truth parts. The authors show that this approach allows for part-aware shape reconstruction, unsupervised segmentation, and shape editing.

The two-step generation approach with geometries and instances and the loss with shape-based part assignments both seem like interesting contributions to me. Using neural representations of part geometries seems a bit less novel, since it has been used before in prior works (for example, Neural Parts, Local Deep Implicit Functions).

**Questions:**

Providing a clear description of what the authors consider to be the main practical contributions of their representation with repeatable parts of existing shape generation methods, and how these contributions are demonstrated in the paper would be good.

**Limitations:**

The authors show some limitations of their method.

**Strengths And Weaknesses:**

Strengths:
- The separation of part generation into geometry and instances seems like an interesting idea to make use of the compositional nature of many shapes, and seems to be relatively novel in the context of part-based shape generation.
- Implementation seems technically solid.
- The paper is generally well-written and easy to read.
- The evaluation shows some advantages in unsupervised shape segmentation.

Weaknesses:
- The authors do not clearly demonstrate the advantages of their contributions. What is the advantage of having a representation with repeatable parts over current work with non-repeatable parts or current work without parts? An more thorough evaluation of some applications that are only possible with repeatable parts, or that clearly perform better than the state of the art would be necessary.
- The evaluation focuses on shape reconstruction, however existing methods like Occupancy Networks perform significantly better on shape reconstruction (Occupancy Networks should be included in the left part of Table 1). There could be other applications where working with repeatable parts is necesary or beneficial (like part editing, part mixing, inductive biases for generation, interpolation, etc.), but these are not demonstrated thoroughly enough. Only one example of shape editing is provided.
- The segmentation experiments are reasonable, but by themselves not sufficient for acceptance, and Table 1 in the supplementary seems to suggest that using repeatable parts actually hurts segmentation performance compared to using non-repeatable parts.

In summary, I like the idea of using repeatable parts to represent shapes, but the evaluation does currently not convincingly demonstrate the advantage of such a representation.

Details:

- Using a part-based representation with part re-use is described as a major advantage of the proposed method in the introduction, and it is a major part of the technical contribution. However, from the evaluation it is unclear what the advantage of such a representation is over the state-of-the-art. For reconstruction accuracy, non-part based representations like OccNet seem to perform better, so that does not seem to be a strong suit of the proposed representation, or possibly of part-based representations in general. The segmentation comparison does show some advantage in Table 1, even if the results are a bit mixed, but Table 1 of the supplementary shows that the segmentation performance is lower with part re-use. There is one edit example in the supplementary that shows some down-stream advantages of part re-use, but additional examples, ideally with more repeated parts (storage furniture?) and possibly mirrored parts (usually the chair arms are mirrored, not related by only translations and rotations) would be more convincing. Also, metrics that specifically measure the success of the part re-use are missing, for example measuring how often symmetric parts in a shape are correctly modeled with the same geometry and how often parts that have different geometry are incorrectly modeled as instances of the same geometry.

- Comparisons to Neural Parts and Local Deep Implicit Functions (LDIF) both seem relevant, since they both perform unsupervised shape decomposition and use neural representations of part geometries. For LDIF, the authors argue that it is not included due to using a local feature encoding compared to the global feature encoding of ProGRIP. What are the specific applications or tasks that LDIF can't do due to its local feature encoding, but that ProGRIP can do? This needs to be discussed and clarified if the goal is to show that an empirical comparisons to LDIF is not needed since there is a clear theoretical advantages. I can imagine, for example, that a single global latent space enables interpolation or generation, which may be more difficult with local latent spaces. Although ideally such an argument is backed up by empirical evidence.

- It might be good to clarify the extent to which the method is unsupervised in the introduction. It is true that the method can be trained on an unannotated dataset, but only if a part annotation is generated for the dataset as a pre-processing step, possibly using existing unsupervised methods. Calling it a fully unsupervised approach in the introduction without further clarification may incorrectly create the expectation that the method is trained end-to-end with only the final occupancy loss as supervision.

- To a lesser extent, calling the representations a shape program may also create the unrealistic expectation of working more complex program. The current representation may better be described as structural representation rather than a program, since only two types of 'operators' are effectively used, always in the same order: first box creation and then instancing. But I would consider fixing this point (for example by changing the title and introduction) optional.

- I agree with the authors that representation compactness is a good measure for a learning-based model. However, it seems like the F-score the authors use measures the reconstruction quality with geometry (the information stored in the z-vectors), but only measures space requirements without geometry (i.e. without counting the information stored in the z-vectors). This seems like an unfair comparison, as the other methods do not make use the z-vectors to store additional information that is useful in improving the F-score.

- The following papers could be added to the related work:
	- SPAGHETTI: Editing Implicit Shapes Through Part Aware Generation, Hertz et al., ArXiv 2022 (concurrent work)
	- SDM-NET: Deep generative network for structured deformable mesh, Yang et al., Siggraph Asia 2019
	- PQ-NET: AGenerative Part Seq2Seq Network for 3D Shapes, Wu et al., CVPR 2020
	- DSG-Net: Learning Disentangled Structure and Geometry for 3D Shape Generation, Yang et al., Siggraph 2022 (concurrent work)
	- Generative 3D Part Assembly via Dynamic Graph Learning, Huang et al., NeurIPS 2020
	- Write, Execute, Assess: Program Synthesis with a REPL, Ellis et al., NeurIPS 2019
	- InverseCSG: Automatic Conversion of 3D Models to CSG Trees, Du et al., TOG 2018
	- Engineering Sketch Generation for Computer-Aided Design, Willis et al., CVPR 2021Workshop paper
	- Learning Adaptive Hierarchical Cuboid Abstractions of 3D Shape Collections, Sun et al., Siggraph Asia 2019
	- ParSeNet: A Parametric Surface Fitting Network for 3D Point Clouds, Sharma et al., ECCV 2020

- Since the box generation model is pre-trained without knowledge of the geometry, it should create two instances of the same box in cases where two boxes are have the same shape. But same box shape does not necessarily mean same geometry (for example, two boxes may contain mirrored geometry, or just geometry that has slightly different details, etc.). Did you observe the model successfully splitting two instances of the same box into two different boxes in the refinement step if the two boxes are the same but not their geometry? Showing such an example might be good to confirm that this is not a problem.

- Transformers typically work on ordered sequences (step i takes the output of step i-1 as input), some details should be given how the transformer is set up here to be order-invariant. For example, does each step still output a probability distribution that is then sampled probabilistically (the standard setup for transformers), or does each step regress/classify output values deterministically (given the input)? If each step samples a potentially multi-modal probability distribution, how are the samples of different steps coordinated to make sure only compatible modes are selected when sampling?

- Allowing for a reflection transformation in the instances seems like it could enable significantly more part re-use, since parts are often mirrored. Was there a specific reason this was not used? Or can the rotation R also be used to model a reflection? This could be discussed in the future work for example, or a clarification is needed that R can also describe reflections.

- The threshold used to compute the F-score metric should be mentioned.

---

> ### Author Response · Authors · 2022-08-02
> **Response to Reviewer JZ3w, thank you for the feedback! (1/2)**
>
> - **Benefit of having repeatable parts**:  Please note that we don’t claim the formulation of repeatable parts as our contribution. Instead, it’s a line of established research [1, 2, 3] that we further by introducing implicit shape representation and our training strategy. For the value of repeatable parts, on a high-level, learning to understand repeatable geometric structures is a big step forward towards intelligent shape understanding. More practically, in addition to showing promising reconstruction and segmentation, the advantages have been demonstrated in more semantic shape editing and compactness (Sec.B&C of our supplementary). As per the reviewers’ proposals, we added more shape editing examples and a shape interpolation application in our updated supplementary materials. We’d also like to acknowledge the application of shape programs with repeatable parts illustrated in prior works, such as shape completion [3], novel shape generation, and directed shape manipulation [2].
>
> - **Use of repeatable parts _hurts_ in segmentation? (Tab. 1 in supp)**:  Using repeatable parts, **helps** and does not hurt in segmentation. We’d like to clarify any factual misunderstanding here.
>     * In Tab.1 (right) of the main paper, we conducted an "instance segmentation” evaluation. This means in the ground-truth shape part labels, we treat copies of the same part (e.g. legs of a chair) as different label predictions. This is the standard protocol adopted by prior works [4, 5]. We observe that ProGRIP does comparable or better across all classes, including CubeSeg, which can be thought of as a non-repeatable abstract shape representation.
>     * In Supp.Tab.1, we also show "semantic segmentation" results. This means in the ground-truth shape part labels, we treat all copies of the same part as a single label. This is measuring the extent to which the detected repeatable parts are semantically similar. This is generally a harder task (one can see it as instance segmentation + classification) hence the lower performance than "instance segmentation" alone. To avoid similar confusion in the future, we update the table header to “instance segmentation” and “semantic segmentation”.
>
> - **Additional edit examples?**: We have added a few more editing examples in our updated supplementary materials. We may also release a demo interface in the future supporting semantic level editing, as we show in our examples. It will only be enabled by our repeatable parts.
>
> - **Measure of part re-use**: while there are no established metrics for this, we made our best efforts computing the portion of labeled part surface points that are within a threshold (0.02) to our reconstructed repeatable part surface. This is indicative of the “how good a semantic part is reconstructed by our repeatable part''. Results are (higher is better):
>     * | Method | chair | table | airplane |
> | --------- | ------- | ------ | -------- |
> | CubeSeg | 46.02 |  36.36 | 66.33 |
> | ProGRIP | **85.35** | **77.49** | **93.47** |
>     * We see from the results that ProGRIP’s repeatable part structure is closer to semantic decomposition of the object compared to CubeSeg.
>
> - **OccNet/LDIF baseline**: Please note that we don’t argue that having local feature encoding is detrimental to ProGRIP. However, due to its architectural complexity, we leave it as an open direction for future research. While there are several methods like OccNet/LDIF that focus solely on high fidelity reconstructions, the key contribution of ProGRIP is a structured shape representation that has favorable properties (such as repetitions, symmetry, interpolation, editing, etc.) while maintaining decent reconstruction fidelity. This is not achievable by prior methods that propose similar representations (Shape2Prog, CubeSeg, BSP-Net). To compensate for the comparison with similar representations as LDIF (gaussian-based and non-repeatable), we have a comparison with SIF [6] in our original supplementary materials, Tab. 4. We find that our geometric fidelity is overall slightly better than SIF while also​​ possessing the property of modeling repeatable structure.
>
> - **Clarify the extent to which the method is unsupervised**: We clarify that unsupervised refers to “matching the oriented bounding boxes of predicted repeatable parts to non-repeatable box-based shape decomposition” (original or updated main paper, L48–49), which is also obtained without annotations.

---

> ### Author Response · Authors · 2022-08-02
> **Response to Reviewer JZ3w, thank you for the feedback! (2/2)**
>
> - **Compactness computation**: Thank you for pointing out that the shape latent should be considered in the compactness measures. Our primary motivation with original Supp.Fig.2 (or updated Supp.Fig.3) was to show that ProGRIP has only a small # of primitives (~5-6) while having decent reconstructions. Having a compact representation aids interpretability and editing. But we agree that a significant portion of gains in the y-axis (F-Score) is by virtue of our z-latents. For a fair comparison, in Tab. 2 and Tab. 3 in our original supplementary materials, we report the F-score of ProGRIP (box, without implicit z-latents) and CubeSeg (w/ implicit), respectively. We observe that ProGRIP (w/ implicit) is a method with high compactness as well as good shape expressivity. We have also updated our supplementary text to clarify that our method has implicit functions contributing to F-Score while others do not (Sec. C).
>
> - **Additional Citations**: Thank you for the relevant pointers! We have updated the main paper citing these works (including concurrent work) in appropriate sections.
>
> - **Distinguishing fine-grained geometric difference with the same box**: In the ShapeNet dataset, we didn't notice such issues. Moreover, distinguishing fine geometry can be done by modifying our hard shape copies (repeating strictly the same shape) to an "as geometrically similar as possible" loss term, such as similar occupancies from implicit functions to guarantee such separation in fine details.
>
> - **Transformers**:  Transformers are also applicable for our unordered set prediction problem. This is inspired by prior works such as DeTR [7] that also use transformers similarly.  We leverage that the self-attention mechanism is permutation invariant and have a decoder that decodes the transformed tokens independently (and in parallel) to predict a set of candidate parts and their existence, trained with a matching loss.
>
> - **Reflection transformation**: We consider the same transformations used in Shape2Prog. The challenge of adding reflection is making it differentiable for backpropagation, which can be an interesting future direction.
>
>
> - **Threshold for F-score computation**: Please see Tab. 1 of our original paper and L43 of our original supplementary materials for the threshold value (0.01).
>
>
>
> [1]  R Kenny Jones, Theresa Barton, Xianghao Xu, Kai Wang, Ellen Jiang, Paul Guerrero, Niloy J Mitra, and Daniel Ritchie. Shapeassembly: Learning to generate programs for 3d shape structure synthesis. In ACM TOG, 2020.
> [2] R Kenny Jones, David Charatan, Paul Guerrero, Niloy J Mitra, and Daniel Ritchie. Shapemod: Macro operation discovery for 3d shape programs. In SIGGRAPH, 2021.
> [3] Yonglong Tian, Andrew Luo, Xingyuan Sun, Kevin Ellis, William T. Freeman, Joshua B. Tenenbaum, and Jiajun Wu. Learning to Infer and Execute 3D Shape Programs. In ICLR, 2019.
> [4] Zhiqin Chen, Andrea Tagliasacchi, and Hao Zhang. Bsp-net: Generating compact meshes via binary space partitioning. In CVPR, 2020.
> [5] Chuhang Zou, Ersin Yumer, Jimei Yang, Duygu Ceylan, and Derek Hoiem. 3D-PRNN: Generating Shape Primitives with Recurrent Neural Networks. In ICCV, 2017.
> [6] Thomas O. Binford. Visual Perception by Computer. Invited talk at IEEE Conf. on Systems and Control, 1971.
> [7] Nicolas Carion, Francisco Massa, Gabriel Synnaeve, Nicolas Usunier, Alexander Kirillov, and Sergey Zagoruyko. End-to-end object detection with transformers. In ECCV, 2020.

---

> > ### Comment · Reviewer_JZ3w · 2022-08-09
> > **Discussion**
> >
> > Thanks to the authors for the clarifications, and sorry for my late response. I am slightly more positive about the paper after the author have clarified some misunderstandings on my part about Tab.1 in the supplementary. I think overall, between the added edit results and the clearer description of Table 1, that actually contains a measure of part-reuse I was asking for in its semantic segmentation results, the paper shows the benefits of working with parts and re-using them convincingly enough. I therefore raise my rating to a borderline accept.
> >
> > There are still many things that should be improved, and I would condition acceptance on the following changes:
> > - Changing Figure 3 of the supplementary to either show the F-score of ProGRIP (box) instead of ProGRIP (implicit), or changing the x-axis of the plots to include the z-latents in the measure of compactness (for example by counting the number of float parameters required to represent an object). In its current form Figure 3 is misleading. When using ProGrip (box) for the F-score, for example, ProGRIP should still show a rasonable advantage in the figure.
> > - Adding SIF (and ideally also OccNets) to the semantic segmentation experiment in Table 1 of the supplementary. The main advantage over methods like SIF, NeuralParts and LDIF seems to be that ProGRIP uses repeatable parts (as also argued by the authors in the related work section). Currently, the semantic segmentation experiment seems to be the main experiment showing the advantage of using repeatable parts quantitatively. Therefore, adding SIF and OccNets to the semantic segmentation experiment would be a good way of showing the main strength of ProGRIP over these two methods.
> > - Clarifying in the introduction to what extent the method is unsupervised, by clearly mentioning that an existing unsupervised decomposition method is used to generate part ground truth. Currently, this does not seem clear enough to me, for example 'we propose an unsupervised learning objective' is a bit misleading, since it is supervised with an existing part decomposition, also 'which allows learning from unannotated shapes' could easily be misinterpreted by the reader in this context.
> >
> > The following changes would greatly improve the paper, but I would not consider them strictly necessary for acceptance:
> > - Adding an ablation that shows the effect of using/not using repeatable parts on reconstruction, instance segmentation, and semantic segmentation. Previously I thought that Table 1 of the supplementary provided part of this ablation, but was corrected by the authors. This ablation seems quite relevant to understand what the advantages of repeatable parts are.
> > - Comparing to Neural Parts and LDIF.

---

> > > ### Author Response · Authors · 2022-08-09
> > > **Thank you**
> > >
> > > Thank you again for taking the time and care to consider both our original manuscript and our responses!  This has certainly made the work stronger and clearer.  We will make the adjustments as you suggested in the revised version, at least to Figure 3 and Table 1 of the supplementary and the introduction.

---

### Official Review · Reviewer_rmbs · 2022-07-12

**Rating:** 7
**Confidence:** 4
**Soundness:** 3 good
**Presentation:** 4 excellent
**Contribution:** 4 excellent

**Summary:**

The paper introduces a novel unsupervised framework for representing repeatable parts as shape programs with part-based implicit functions. The authors propose to utilize implicit functions to represent parts, which improve the representation capacity of shape programs. Meanwhile, a matching-based unsupervised learning objective is devised to learn the shape program from unannotated shapes. Experimental results show the effectiveness of the proposed method in shape reconstruction fidelity and semantic segmentation for chairs, tables and airplanes of the ShapeNet dataset.

**Questions:**

Please see the weaknesses.

**Limitations:**

Please see the weaknesses.

**Strengths And Weaknesses:**

Strengths:
+ Both the problem addressed in the paper and its proposed solutions are innovative and interesting.
+ The proposed matching-based unsupervised training objective is sound.
+ The experiment shows quite marginal performance improvement.
+ The paper is well-organized and easy to follow.

Weaknesses:
I don't really have any major concerns; some minor concerns are as follows:
1. Since part-based implicit shape representation is used, it would be better to compare the training and testing complexity between the proposed method and the baseline, such as BSP-NET.
2. Does the proposed method require a canonical orientation of shapes.
3. the authors should provide qualitative and quantitative results on more categories, such as car and lamp.
4. It would be good if the authors could conduct more experiments to show the expressiveness of the new shape representation. For example:
i) It would be interesting to explore the feature space to validate if the method can accomplish smooth interpolation and extrapolation of shapes.
ii) Can the new shape programs be adapted for single-view 3D reconstruction?
5. Are the trained models category-specific or a single universal model is trained for these 3 categories? How do you make sure the comparison is fair in Tab. 1. All baselines are trained in the same experimental setting.

---

> ### Author Response · Authors · 2022-08-02
> **Response to Reviewer rmbs, thank you for the feedback!**
>
> * **Comparing with BSP-NET on training and testing complexity**: Below are the statistics of training and testing complexity for ProGRIP and BSP-Net:
>     * | Method     | BSP-Net | ProGRIP |
>       | ----------- | ----------- | ----------- |
>       | Training |  ~7d10hr  | ~2d1hr  |
>       | Test   | ~1.27s/mesh  | ~1.35s/mesh  |
>     * We train BSP-Net as specified in the paper using the authors’ open-source code release. They use hierarchical training (16^3 → 32^3 → 64^3) for the continuous domain, which takes a total of ~4d15h, followed by a 64^3 discrete stage which takes an additional ~2d20h. All experiments were run on a Titan RTX GPU, similar hardware to ProGRIP experiments.
>     * From the statistics, we can see that ProGRIP is faster than BSP-Net during the training stage and comparable to BSP-Net on test time inference.
>
> - **Canonical Orientation Requirement**: Our method doesn’t require any canonicalization of object shapes in the data itself.
>
> - **Experiments on more categories**: Please note that we select the Chair and Table classes to compare with Shape2Prog, which can only work on these classes. We further select the airplane class as a representative demonstration that our ProGRIP method applies to any class. Due to the limited time of the rebuttal period, we are not able to finish the experiments for categories such as cars and lamps, in particular, due to the data processing and the training of baselines (e.g., 7 days+ for BSP-Net). We will include the results for these classes as soon as possible once the experiments finish.
>
> - **Expressiveness of the new shape representation**:
>     * Interpolation: We thank the reviewer for the proposal. Please see the interpolation video (.mp4) in our updated supplementary materials.
>     * Single-view 3D reconstruction: Our ProGRIP can be adapted to single-view 3D reconstruction when combined with a proper single-view encoder. This is a quite interesting future direction yet beyond the scope of our current work.
>
> - **Is ProGRIP category-specific?**: Yes, ProGRIP is focused on modeling a single class.
>
> - **How to ensure fair comparison in Tab. 1?**: We use the exact same task set up to ensure this, including using the same train/eval data split, using the same set of sample points for training all different implicit shape based methods, and setting identical hyperparameters (specifically, the number of samples for computing the metrics and the threshold used in F-Score computation) for evaluation.

---

> > ### Comment · Reviewer_rmbs · 2022-08-10
> > **Rebuttal Acknowledgement**
> >
> > Thank you for the answers. I believe that most concerns have been properly addressed. I am inclined to stick to my original rating.

---

### Author Response · Authors · 2022-08-02
**Submission Updates Record**

* Main Paper:
    * Added all suggested references to the main paper.
* Supplementary Materials:
    * Added a video of interpolation (.mp4 in the zip file).
    * Added additional editing examples as a figure and a sentence inline referring to the figure.
    * Fixed a typo in Supplementary Tab. 1.
    * Clarified the instance segmentation vs. semantic segmentation experiment in supplementary materials (Tab. 1 and Sec. D).
    * Clarified the use of the z latent feature in our compactness experiment (Sec. C) as suggested by R#2.

---

### Author Response · Authors · 2022-08-08
**Looking forward to your response**

Dear Reviewers,

Thank you again for your constructive reviews, which have helped us improve the quality and clarity of the paper.   We have updated the individual response to each of your reviews under your thread, and have supplied a list that summarizes the changes.   We hope that we have been able to address your concerns.  As we approach the end of the discussion period approach, please don’t hesitate to let us know if you have any additional questions or comments.  We look forward to the discussion!

Thanks for your time,

Authors

---

### Meta-Review · Area_Chair_11r8 · 2022-08-28

**Recommendation:** Accept
**Confidence:** Certain

**Metareview:**

All reviewers recommend acceptance of this paper. They find the approach of repeatable parts innovative and the paper well written. The AC concurs

**Award:**

No

---

### Decision · Program_Chairs · 2022-09-14

Accept